# SADE: A SCENE-TEXT AUTOREGRESSIVE DIFFUSION ENGINE FOR CHARACTER SEQUENCE RECOGNITION

## ABSTRACT

We consider the problem of training an optical character recognition (OCR) model to read short alphanumeric scene-text sequences, such as number plates or vehicle type labels, in scenarios where labelled training images are limited in quantity and sequence variety. OCR models may under-perform in these scenarios, so we explore whether a diffusion model can be trained on the small set of labelled images, to generate synthetic images with similar background statistics but new character sequences. We find that a diffusion model struggles to generate characters in positions of the sequence where they did not appear during training. We address this problem by introducing SADE: a scene-text autoregressive diffusion engine that generates multiple image parts one by one, conditioned on previously generated image parts for visual coherency. This approach reduces the effective number of possible positions for a character, and increases the diffusion model's ability to generate characters in novel positions of the full sequence. Our results indicate that SADE can indeed lead to substantial improvements in OCR accuracy in data-scare scenarios, particularly on sequences with characters at positions not encountered in the original training data.

## 1 INTRODUCTION

Optical character recognition (OCR) is the computer vision task of extracting and identifying text from an input image. For applications such as document digitization, there is often an abundance of training data and the text might be relatively easy to extract and interpret. For scenarios involving scene-text captured in less constrained environments, and where training data is limited in quantity and variety, OCR performance can degrade substantially. In such scenarios, an OCR model may struggle to recognize characters appearing in positions that were infrequently or never encountered in the training data, posing limitations on the model's ability to generalize to new sequences not seen during training. A practical example is when a motor vehicle manufacturer uses OCR to validate the type label on the rear of the vehicle. The introduction of a new vehicle type may render the model unreliable until additional training data is collected. Off-the-shelf pretrained OCR models, while effective for general OCR tasks involving natural language, might fail to meet the required performance of specialized scenarios like vehicle type label recognition, where character-level accuracy is vital.

A common trend in the literature to achieving state-of-the-art performance on OCR tasks is to take a data-centric approach (Liao et al., 2019; Yim et al., 2021; Zhu et al., 2023), rather than improving the OCR technology itself. In line with this perspective, we explore the use of a synthetic data generator to bridge the performance gap of OCR for sequences seen during training and those with characters in positions infrequently or never seen during training. An off-the-shelf generative model is not suited for accurate generation of specialized scene-text character sequences, and a standard diffusion model trained on a limited dataset is unable to generate characters in new positions of the sequence. These findings lead us to introduce SADE: a scene-text autoregressive diffusion engine that produces realistic scene-text images of sequences both seen and unseen in training. SADE simplifies the generation process by progressively generating partial images of subsequences, each conditioned on a previously generated part for visual coherency in the full image.

To evaluate SADE's ability to generate synthetic data that closely resembles the original training data, we experiment first with a mock number plate dataset of rendered images, where we can con-

trol the quantity and characteristics of the training data, and then also with a real-world dataset of images of vehicle type labels. We employ OCR as a proxy to measure performance improvements achievable from the use of synthetic data generated by SADE. Our main contributions can be summarized as follows:

- We introduce an autoregressive technique for progressively generating image parts through a diffusion process, that form a coherent whole image.
- We show that our approach can generate realistic scene-text images, specifically of sequences containing characters in positions not encountered in the original training data, and that these generated images facilitate substantial improvements in OCR accuracy.

## 2 RELATED WORK

While there are ongoing efforts to enhance OCR model technology (Wang et al., 2024; Buoy et al., 2024; Zhang et al., 2024b; Chi et al., 2024), there is also a focus on improving scene-text OCR performance by developing datasets with sufficient variation in backgrounds, fonts, distortions, and noise. Collecting and annotating real-world images is often costly and time-consuming, so a common approach is to generate fully annotated synthetic datasets.

### 2.1 RULE-BASED APPROACHES

Jaderberg et al. (2014) developed a widely used synthetic scene-text recognition dataset called MJ, using a six-stage rule-based engine. A dataset commonly used alongside MJ is ST (Gupta et al., 2016), built from the SynthText engine. Zhan et al. (2018) expanded on SynthText by identifying regions where text would naturally appear. Yim et al. (2021) introduced two text selection strategies to address misrepresentations of text distributions, and developed a five-stage scene-text generation pipeline called SynthTIGER. The widespread use of these datasets in recent work on OCR (Buoy et al., 2024; Fujitake, 2023; Du et al., 2024; Liu et al., 2024) highlights the value of synthetic data but also reveals a possible need for more modern and realistic scene-text generation engines. To that end, we investigate the efficacy of diffusion-based image generation.

Generating scenes in virtual 3D space enables more accurate perspective transformations and lighting, compared to 2D image editing. SynthText3D (Liao et al., 2019) embeds text into virtual environments and renders images with varied illumination and camera angles. Long & Yao (2020) addressed the scaling and diversity limitations of SynthText3D by leveraging 3D object properties, such as meshes, in their three-stage pipeline called UnrealText. Despite the realism of this synthetic data, 2D engines seem to remain the more popular choice for training and benchmarking OCR.

### 2.2 LEARNING-BASED APPROACHES

Rule-based methods can be efficient for generating large datasets, but they struggle to capture certain natural image details. Learning-based approaches offer an alternative by automatically incorporating more realistic variations. Storchan & Beauschene (2019) used a generative adversarial network (GAN) in an end-to-end OCR system, achieving strong results for reading damaged faxes and PDFs. Li et al. (2023) employed adversarial training to augment characters in tail classes, setting a new state-of-the-art in oracle bone script recognition. Similarly, Yeleussinov et al. (2023) demonstrated the effectiveness of GANs in boosting Kazakh handwritten text recognition accuracy.

The latest advancements in image generation leverage denoising diffusion probabilistic models (Ho et al., 2020). Zhu et al. (2023) introduced conditional text image generation with diffusion models (CTIG-DM), and showed improved performance of existing text recognizers and the generation of images of out-of-vocabulary words. Zhang et al. (2024a) addressed the challenge of generating multilingual scene-text images with their framework called Diff-Text, achieving improved accuracy and normalized edit distance from OCR tools. While these studies highlight the benefits of generative models for scene-text recognition, our work in this paper focuses on the recognition of short alphanumeric text sequences where character-level accuracy is vital and training data is limited in quantity and variety.

## 3 METHODOLOGY

Our interest is in training an OCR model to read short alphanumeric sequences (like number plates, or vehicle type labels) in images. We assume access to a labelled training set of images, cropped roughly around the sequences, that might be limited in size and sequence variation. The idea is to train a diffusion model to generate additional images with similar background statistics but new character sequences, to assist the OCR model.

### 3.1 CHARACTER SEQUENCE CONDITIONING

Diffusion models are often conditioned on encodings from a large language model (LLM) (Rombach et al., 2022; Saharia et al., 2022; Zhang et al., 2024a), but because LLMs are designed to capture the meaning of tokens by the presence or absence of other tokens, and are to some extent insensitive to token order, such encodings are unsuitable in the case of nonsensical sequences where character-level accuracy and order are important.

Our approach to condition a diffusion model on a sequence involves creating character-level embeddings and concatenating them to form a sequence embedding. This allows the diffusion model to identify a character within a sub-space of the sequence embedding and infer its position.

### 3.2 GENERATING CHARACTERS IN UNSEEN POSITIONS

For a diffusion model to correctly identify characters and their positions from the concatenated character embeddings, it must see each character in each position during training. If the model does not see a certain character in a particular position during training, it will not recognize the embedding vector at that position and fail to generate the character.



This behaviour can be demonstrated in a small setup where a diffusion model is trained on simple rendered images containing all possible 3-character sequences of $\{$a, l, m, q, x, 2, 3, 4, 5, 6$\}$ that do not have the letter l in the third position, in a clear black font on a grey background. Figure 1 shows samples generated by the model, when conditioned on sequences with l in the third position. The model fails, despite having seen the character in other positions.

Figure 1: Samples conditioned on the sequences shown above each, from a diffusion model trained on sequences without an l in the third position.

### 3.3 OVERLAPPING IMAGE PARTS

One approach to increase the frequency of characters appearing in different positions is to split a sequence, thereby decreasing the number of positions. For example, ABCD can be split into AB and CD, so that characters in the first and third positions of the full sequence now both appear in the first position of a subsequence. Images can be split vertically into parts, and a diffusion model can be trained to generate such image parts individually. However, processing image parts independently prevents the model from applying consistent orientation and background details across the parts. We therefore include the previously generated part (from left to right) as an additional channel in the input image, thereby creating a sort of autoregressive process. In this way, the diffusion model has reference to the orientation and background that should be applied to the part being generated.

Generated image parts can be concatenated to form a final whole image, but we found that this can easily lead to unwanted artefacts at part borders (see Appendix A for examples). Splitting an image vertically can result in character pieces being cropped into the wrong image part. Since the model is not conditioned on the embedding of such cropped characters, it interprets them as background. The solution we propose is to split an image with an overlap of at least one character. Figure 2 illustrates how an image (from our mock number plate set) is split into seven equal-width pieces and combined into two overlapping parts. While this approach does not guarantee that a piece always contains a single or whole character, it produces roughly the correct number of characters per image part.

We call the method SADE: Scene-text Autoregressive Diffusion Engine.

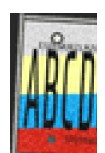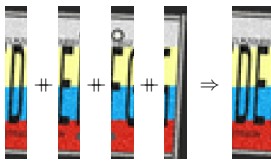

Figure 2: Splitting a 7-character image into two image parts, with a single character overlap.

## 3.4 TRAINING AND SAMPLING FROM SADE

To guide SADE in determining the orientation and background to apply to an image part, we stack the previous image part as an additional channel of the image for which noise is being predicted. A black image is used for the first image part. The training process is detailed in Algorithm 1, which is an adaptation of the training algorithm by Ho et al. (2020). The data distribution $q$ consists of training images with corresponding character sequence labels. The function *Split* vertically divides an image into $N$ parts with the appropriate amount of overlap, while *ApplyNoise* adds noise to an image according to the DDIM beta scheduler (Song et al., 2021). $\mathbf{X}_{i,t}$ and $y_i$ denote the $i$th image part and label of image $\mathbf{X}_t$, where $t$ is the timestep to which the image is noised, and the black image $\mathbf{X}_{0,0}$ has the same dimensions as the other image parts. The model to be trained is $v_\theta$ with sequence embedding layers $\tau_\theta$, and the loss is measured against the prediction $v_t$ as proposed by Salimans & Ho (2022).

Algorithm 2 details the sampling process for SADE, again adapted from Ho et al. (2020). Here the function *Split* divides a character sequence into parts (with appropriate overlap) corresponding to the $N$ image parts to be generated. The algorithm denoises an image according a noise schedule $\alpha_t$ and $\sigma_t$. Starting with a black image part $\mathbf{X}_{0,0}$ gives the model freedom to generate any orientation and background that will propagate through the rest of the image. The function *Join* combines image parts by pasting over overlapping regions to form a single image.

---

**Algorithm 1** Training SADE

**for** each training step **do**
    $(\mathbf{X}_0, y) \sim q(\mathbf{X}_0, y)$
    $t \sim \text{Uniform}(\{1, \ldots, T\})$
    $i \sim \text{Uniform}(\{1, \ldots, N\})$
    $\{(\mathbf{X}_{0,0}, y_0), (\mathbf{X}_{1,0}, y_1), \ldots, (\mathbf{X}_{N,0}, y_N)\}$
        $\leftarrow \{(\mathbf{0}, \mathbf{0}), Split(\mathbf{X}_0, y)\}$
    $\epsilon \sim \mathcal{N}(\mathbf{0}, \mathbf{I})$
    $\mathbf{Z}_{i,t} \leftarrow [ApplyNoise(\mathbf{X}_{i,0}, \epsilon, t), \mathbf{X}_{(i-1),0}]$
    Take gradient descent step on
        $\nabla_\theta \| v_t - v_\theta(\mathbf{Z}_{i,t}, \tau_\theta(y_i), t) \|_2^2$
**end for**

---

**Algorithm 2** Sampling from SADE

**Require:** a character sequence $y$
    $\{y_1, \ldots, y_N\} \leftarrow Split(y)$
    $\mathbf{X}_{0,0} \leftarrow \mathbf{0}$
    **for** $i = 1, \ldots, N$ **do**
        $\mathbf{X}_{i,T} \sim \mathcal{N}(\mathbf{0}, \mathbf{I})$
        **for** $t = T, \ldots, 1$ **do**
            $\mathbf{Z}_{i,t} \leftarrow [\mathbf{X}_{i,t}, \mathbf{X}_{(i-1),0}]$
            $\mathbf{X}_{i,t-1} \leftarrow \alpha_t \mathbf{X}_{i,t} - \sigma_t v_\theta(\mathbf{Z}_{i,t}, \tau_\theta(y_i), t)$
        **end for**
    **end for**
    $\mathbf{X} \leftarrow Join(\{\mathbf{X}_{1,0}, \ldots, \mathbf{X}_{N,0}\})$

---

## 3.5 EVALUATING A DIFFUSION MODEL WITH AN OCR MODEL

While OCR is a downstream task that can benefit from additional training images, it can also serve as a tool for evaluating the performance of a diffusion model in generating scene-text images. The test performance of an OCR model can be viewed as a measure of how much information about the true data distribution is available to the OCR model in the training data, or to what extent the training data explains the variation in the true distribution. From this perspective, the test performance of an OCR model trained on original data (as opposed to synthetic data) can serve as a baseline metric for a given dataset. Comparing this baseline to the performance of an OCR model trained on synthetic data can thus illustrate whether the synthetic data captures more variation or deviates from the true distribution. We will use three separately trained OCR models to evaluate the performance of a single diffusion model: the first is trained on original data, the second on synthetic data from the diffusion model, and the third on a combination of original and synthetic data.

## 4 EMPIRICAL PROCEDURE

We conduct two sets of experiments to evaluate the ability of SADE to generate images of character sequences. The purpose of the first set is to establish some basic expectations and comparisons between using synthetic data verse the original data for training an OCR model. For control over variation within the data, we introduce a mock number plate dataset. In the second set of experiments, we evaluate SADE on a more complex real-world dataset of vehicle type labels.

### 4.1 MOCK NUMBER PLATE DATA

Our mock dataset is loosely inspired by Venezuelan number plates[1]. A few example images from our set are shown in the top row of Figure 3. We form 7-character sequences from the set {A, L, M, Q, X, 2, 3, 4, 5, 6}, render such a sequence on a plate design, apply random noise, rotation and translation perturbations, and pick a background shade at random. We also simulate poor image quality by scaling down and up again. All images are rendered at a resolution of $112 \times 112$, to be easily divisible into seven character parts.

We construct six separate training sets. The first five contain 40, 50, 70, 80 and 100 sequences, respectively, sampled randomly from the $10^7$ possibilities. For each sequence, five images are rendered with different perturbations. The sixth training set consists of sequences that do not have a 5 in the sixth position, to assess the ability of SADE to generate characters in positions not encountered during training. Starting with an initial set of 15,000 randomly sampled sequences, we remove 1,508 that contain a 5 in the sixth position and reserve those for a test set (to be generated by SADE, after training). A separate test set of 5,000 randomly sampled sequences is also created.

### 4.2 REAL-WORLD VEHICLE TYPE DATA

We demonstrate a possible real-world use case for a motor vehicle manufacturer, by applying SADE to images of vehicle type labels. Starting with an initial set of 7,974 images of 31 unique vehicle type labels, we augment the dataset by randomly occluding single characters. To occlude a character at the beginning or end of a sequence, the image is cropped. When occluding a character in the middle of a sequence, a background region of the image is pasted over the character, and a space character replaces it in the corresponding sequence. These augmentations result in a dataset of 15,382 images with 128 unique sequences.

SADE is designed for fixed-length sequences, while the vehicle type labels vary in length. To accommodate this, we add space characters randomly to the start or end of a sequence until it reaches a desired length. Adding a space to the start has the added benefit of shifting characters, leading to more diversity of characters at each position. Padding is added to the corresponding image to account for the extra characters, with the colour of the left- or right-most pixel halfway up the image. Images are resized to a resolution of $120 \times 120$ for divisibility by 3, 4 and 5 (the possible sequence lengths). Examples of images from this dataset are shown in the bottom row of Figure 3.

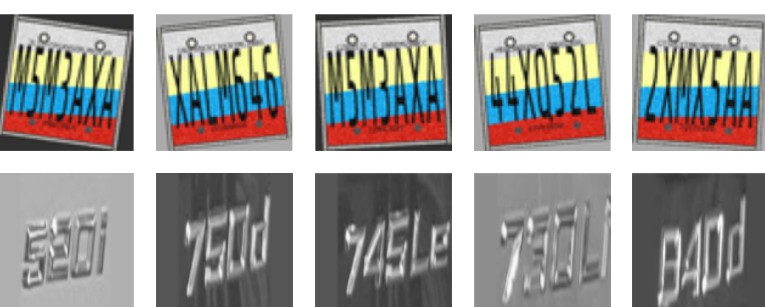

Figure 3: Example images from our mock number plate dataset (top row), and the real-world vehicle type label dataset (bottom row).

---

[1] http://www.worldlicenseplates.com/world/SA_VENE.html

We create five random splits of the dataset, and will further evaluate performance on a hold-out set of sequences (set 6). The hold-out set includes images of `750d`, `730i`, and `545e`, along with their unique augmentations: `75␣d` and `73␣i`. These augmentations are unique to the hold-out sequences, in the sense that they cannot be derived from any other sequence. For each of the first five datasets, three sequences are reserved for dataset-specific testing. Of the remaining sequences, 5% are used for validation, and the rest for training. Since the number of images per sequence vary, the training, validation and test sets differ in size across the five splits. Table 1 provides a summary. Note that we create a test set for each of the five splits, that is actually used within the broader context of cross-validation across different random splits of the full dataset. The sequences in set 6 are kept completely out of this cross-validation, for final testing.

A seventh set is created to assess SADE's ability to generate characters in novel positions, in a real-world setting. From the initial 31 unique vehicle type labels, `745Le` is the only one with an `e` in the fifth position. As a result, this sequence and its derivatives are reserved for testing in set 7. The remaining sequences are split further for training and validation.

Table 1: Train, validation and test sizes and test sequences for each split of the vehicle type data.

| Set | Test sequences | Train | Val. | Test |
|---|---|---|---|---|
| 1 | `530i`, `75␣i`, `740d` | 10,955 | 577 | 673 |
| 2 | `745Le`, `7␣0Ld`, `50Ld` | 11,441 | 603 | 162 |
| 3 | `40Ld`, `54␣i`, `8␣4i` | 11,561 | 610 | 36 |
| 4 | `750Li`, `7␣0d`, `760Li` | 11,490 | 605 | 111 |
| 5 | `750i`, `74␣d`, `53␣i` | 11,305 | 596 | 304 |
| 6 | `750d`, `730i`, `545e`, `75␣d`, `73␣i` | - | - | 53 |
| 7 | `745Le`, `745L`, `745␣e`, `74␣Le`, `7␣5Le`, `45Le` | 11,413 | 601 | 244 |

### 4.3 DIFFUSION MODEL IMPLEMENTATION

For each mock number plate set, SADE is trained for 12,000 iterations, using AdamW (Loshchilov & Hutter, 2019) to optimize the 'SNR+1' weighting loss (Salimans & Ho, 2022), a learning rate of 0.0006, a weight decay of 0.002, a batch size of 24, a cosine learning rate scheduler that includes warm-up of 200 iterations, and the DDIM noise scheduler with a cosine beta schedule (Nichol & Dhariwal, 2021). The number of denoising timesteps is set to $T = 1000$, the character embedding length is 64, and images are split into three parts. These parameters were selected based on searches and early experimentation. The OCR models accept greyscale images, so for computational efficiency, training images for SADE are also converted to greyscale. Our architecture and training scripts are publicly available[2] and are based on the Diffusers library from Hugging Face[3]. Hyperparameters used in the U-Net architecture are listed in Appendix B.

In the case of the vehicle type labels, the number of image parts and the chosen sequence length are important hyperparameters that determine the number of characters per image part. It might seem ideal for a model to see one character per image part, allowing it to learn all characters in a single position. However, due to the autoregressive nature of our approach, an accumulation of errors over many image parts can lead to distorted characters and poor image quality (see Appendix A for examples). To avoid this, the number of parts and the fixed sequence length must be tuned. For our vehicle type dataset, we found an optimal number of image parts and fixed sequence length to be 2 and 6, respectively, where image parts have one character overlap. Details of the search can be found in Appendix C. We train SADE on the vehicle type sets with a batch size of 16, for 10,000 iterations. All other hyperparameters are the same as those used for the mock data.

In the case of the mock data, we sample 5,000 images in 20 timesteps with each trained model, using sequences randomly selected with replacement. In the case of the vehicle type data, we sample 10,000 images with each trained model. Here we have only 128 possible sequences, meaning that the model may occasionally generate an image of a test sequence. This highlights a key advantage of using a generative model for synthesizing scene-text images for OCR model training.

---

[2]`https://anonymous.4open.science/r/sade-3012`
[3]`https://huggingface.co/docs/diffusers/en/index`

### 4.4 OCR MODEL IMPLEMENTATION

For each diffusion model we train three OCR models: on the original data, on synthetic data, and on a combination of the original and synthetic data at a ratio of 1:3. The framework we use for training and testing these models was introduced by Baek et al. (2019). Each OCR model is trained for 5,000 iterations with a batch size of 128 and validation every 500 iterations. The checkpoint with highest validation accuracy is saved as the final model. No additional data transformations are done, VGG is used for feature extraction, a BiLSTM is employed for sequence modelling, and an attention-based model is used for prediction (Shi et al., 2016). All other training parameters are kept at their default values, and the model is trained end-to-end from scratch. We found more advanced features available in the framework to quickly overfit to our relatively small datasets.

Additionally, we compare our trained OCR models with the pretrained TPS-ResNet-BiLSTM-CTC model provided with the OCR framework by Baek et al. (2019). This model was trained on MJ (Jaderberg et al., 2014) and ST (Gupta et al., 2016), and serves as an off-the-shelf benchmark.

## 5 RESULTS

### 5.1 MOCK NUMBER PLATES

Figure 4 shows typical samples generated by SADE trained on mock number plate images. Image parts are generated with consistent orientation and background, and combine seamlessly. The six images on the left were sampled from a model trained on 100 sequences, and the six on the right from a model trained on sequences where a 5 never appears in the sixth position. We see that SADE is capable of generating sequences containing characters in positions not seen during training. There are a few errors that we will mention again in Section 5.3.

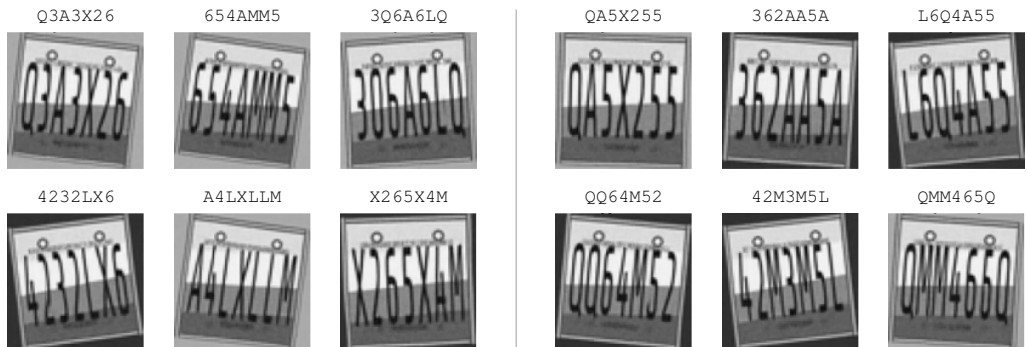

Figure 4: Samples generated by SADE trained on 100 sequences (left), and on sequences where a 5 never appears in the sixth position. Sampling was conditioned on the randomly selected sequence shown above each image, and these sequences were not (necessarily) present in the training set.

Figure 5 illustrates test accuracies from the three OCR models trained for each experiment using the mock data. The consistently high accuracies for models trained on synthetic data indicate how well this data approximates the real distribution. The experiments conducted with 40 and 50 initial training sequences highlight how SADE can extrapolate in data-scarce situations. These results are given in table form in Appendix D.

For experiment 6, where a 5 is never seen in the sixth position, the OCR model trained on the original data achieves a test

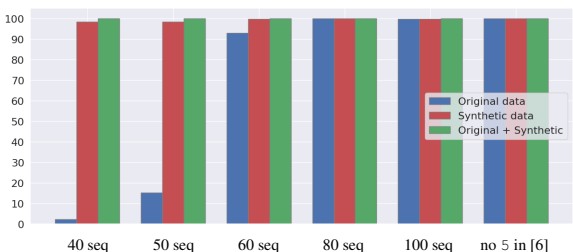

Figure 5: Test accuracies from OCR models trained on combinations of original and synthetic mock number plate images, for six different SADE training sets.

accuracy of 99.96%, misclassifying only `Q66XX5X` and `3A44556`. When tested on the 1,508 images removed from the training set, it misclassifies `666665X`, revealing a weakness in recognizing a `5` in the sixth position. In contrast, the OCR model trained solely on synthetic data misclassifies only `42Q4LA6` in the main test set. Both models trained with synthetic data achieve 100% accuracy on the 1,508 images that were removed from the training set.

## 5.2 REAL-WORLD VEHICLE TYPE LABELS

Figure 6 illustrates test accuracies from the three OCR models trained for dataset splits 1 to 5 of the vehicle type labels (refer to Table 1 for detail on the splits). The left graph shows OCR performance on the corresponding test set of each split, while the right graph shows the performance of all OCR models on the hold-out set (set 6 in Table 1). The OCR models trained only on original data (blue bars) give a mixture of high and low accuracies, with an average of 37.93% and standard deviation of 29.03 on the test sets of the splits, and an average of 78.87% and standard deviation of 17.47 on the hold-out set. The high variance is likely due to some test sets being more similar to the training data than others. The average validation accuracy after training is 99.93% with a standard deviation of 0.09, suggesting that the OCR models are not generalizing well to unseen sequences.

The OCR models trained solely on synthetic data generated by SADE (red bars) also vary, achieving an average accuracy of 68.87% and standard deviation of 33.84 on the test sets of the splits, and an average accuracy of 70.19% and standard deviation of 23.09 on the hold-out set. Figure 7 shows typical samples from the diffusion model trained on set 5 of the vehicle type data. While some samples contain deformed or incorrect characters, the model is able to generate good images of sequences not seen during training (such as the right-most image in the top row of the figure).

The OCR models trained on original and synthetic data (green bars) achieve consistently high accuracy on both seen and unseen sequences, with an average of 97.95% and standard deviation of 2.94 on the test sets of the splits, and an average of 94.72% and standard deviation of 4.09 on the

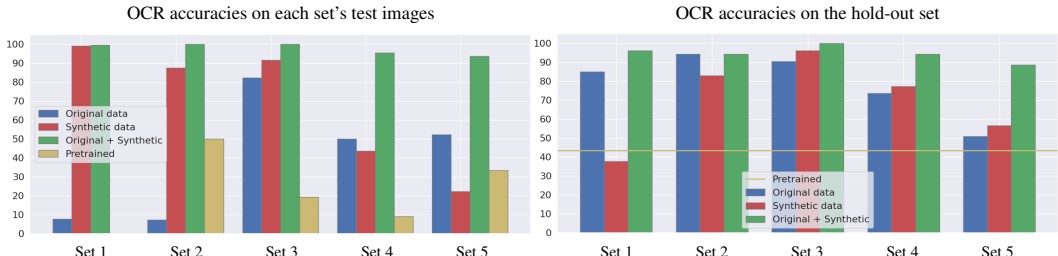

Figure 6: Test accuracies from OCR models trained on combinations of original and synthetic vehicle type label images, for five different SADE training sets, on the test set corresponding to each training set (left) and on the separate hold-out set (right).

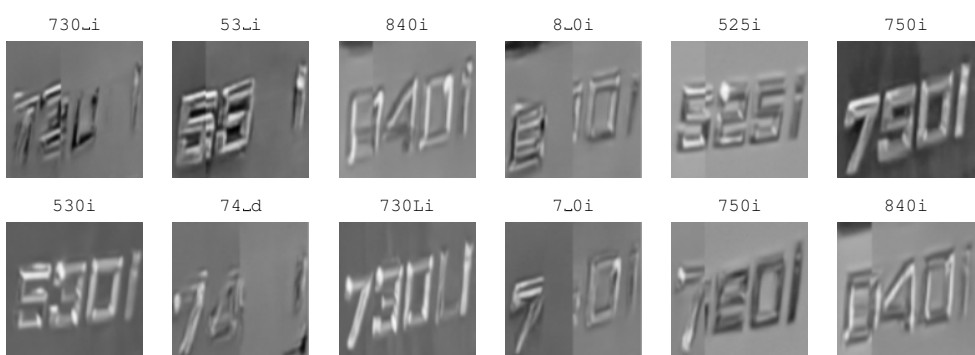

Figure 7: Representative samples generated by SADE trained on set 5 of the vehicle type data. Sampling was conditioned on the sequence shown above each.

hold-out set. This may be attributed to the complementary strengths of the original and synthetic data. SADE can extrapolate and produce unseen sequences, adding greater variation to the training data and preventing overfitting. However, as seen in Figure 7, synthetic data may contain deformed characters that could hinder OCR performance. Through a combination of original and synthetic data, an OCR model can learn precise character representations and generalize to new sequences.

Figure 6 also illustrates (in yellow) the performance of the pretrained TPS-ResNet-BiLSTM-CTC model of Baek et al. (2019) on our various test sets. The model does not perform well on this data, underscoring the necessity for a more specialized solution.

As a final experiment, we consider a split of the data where the sequence `545Le` and its unique augmentations are reserved for testing (set 7 in Table 1). The OCR model trained on the original data is unable to correctly recognise any of the test images, the OCR model trained on synthetic data achieves an accuracy of 72.92%, and the OCR model trained on both original and synthetic data achieves an accuracy of 94.67%. This highlights the advantage of using SADE to generate synthetic data for OCR training in data-scarce scenarios, particularly for sequences where characters appear in positions not encountered in the original data.

### 5.3 LIMITATIONS

While Figures 4 and 7 demonstrate SADE's ability to generate realistic images, they also reveal that high quality is not guaranteed. For example, in the third image of the bottom row in Figure 4, the `4` has an extra leg, and in the right-most image of the bottom row, the `5` looks more like a `6`. In Figure 7, the second image of the top row displays two `5`'s but the second should be a `3`. Although OCR can still benefit from using this data, such deformations limit performance. A possible explanation for the artefacts is that the diffusion model relies too much on copying the right side of the previous image part into the next, rather than using the sequence embedding to guide character generation.

While SADE can generate characters in novel positions, its effectiveness is dependent on the training data and certain situations can pose challenges. For example, if a character appears only in the first position of a sequence during training, the sequence must be split with either one or very few characters per subsequence to enable the generation of that character in other positions. Depending on the total sequence length, the cumulative error from having many subsequences may hinder the generation of realistic images (as we demonstrate in Appendix A). Therefore, the number of image parts and the fixed sequence length should be tuned according to the characteristics of the dataset.

## 6 CONCLUSION

We identified and addressed the challenge that OCR models face when training on images of short alphanumeric scene-text sequences, in data-scarce scenarios and where some characters do not occur in certain positions. Our goal was to create synthetic data with sufficient variation in order to improve the training of an OCR model. To achieve this, we developed SADE, a novel autoregressive diffusion engine capable of generating images of new sequences, including those with characters in positions not seen in the original training data.

We experimented with a mock number plate dataset to showcase SADE's ability to generate characters in new positions and produce data that resembles the original data. We also demonstrated SADE's practical value on a real-world dataset of motor vehicle type labels, by investigating the benefits of synthetic data for training an OCR model to recognize vehicle type labels.

For future work we plan to explore pretraining an OCR model on synthetic data and fine-tuning it on real data. We also aim to reduce unwanted artefacts by providing SADE with less visual context from the previous image part and more textual context for the current one.

### REPRODUCIBILITY

We provide details of each training and sampling setup in the main body of the paper, and the hyperparameters for each model in Appendix B. The code for training and sampling a diffusion model using our methods is available at: `https://anonymous.4open.science/r/sade-3012/README.md`

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

## A  ERRORS AND LIMITATIONS

In Section 3.3 we mentioned that training a diffusion model to generate separate image parts with no overlap can lead to unwanted artefacts. Figure 8 shows samples from such a model that generates an image in two parts of 4-character sequences each (the full sequences have seven characters, so images are padded with an extra blank piece). Artefacts can be seen on the boundaries between the fourth and fifth characters.

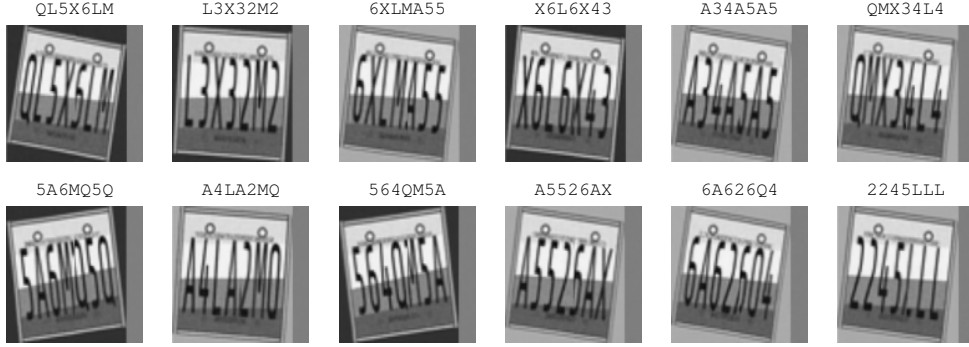

Figure 8: Samples from a diffusion model trained to generate two images parts without overlap, exhibiting unwanted artefacts between characters at positions 4 and 5.

In Section 4.3 we mentioned that splitting the generation process into too many image parts can lead to an accumulation of errors that may distort characters. Examples of this behaviour are shown in Figure 9.

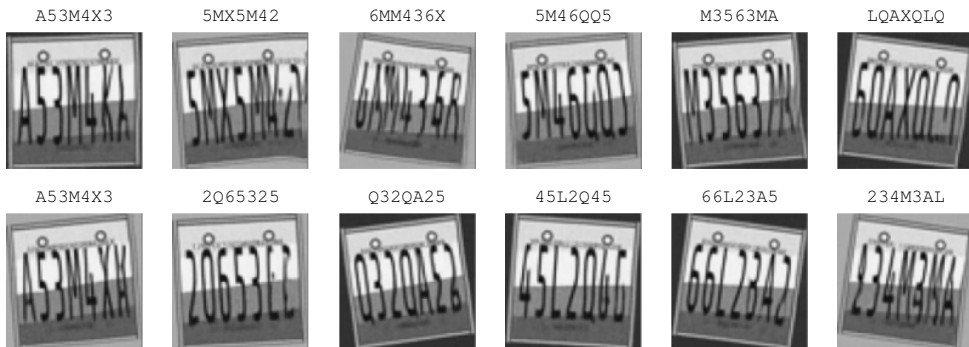

Figure 9: Examples of images generated from seven image parts, demonstrating how errors can accumulate and lead to distorted characters.

## B  U-NET HYPERPARAMETERS

Our diffusion models are based on the Diffusers library from Hugging Face. Non-default hyperparameters used in the UNet2DConditionModel architecture are listed in Table 2.

Table 2: U-Net hyperparameters used in this paper.

| Parameter | Value |
|---|---|
| attention_head_dim | 16 |
| block_out_channels | [256, 384, 384, 448] |
| char_embed_dim | 64 |
| cross_attention_dim | 256 |
| down_block_types | [ DownBlock2D, CrossAttnDownBlock2D × 3 ] |
| encoder_hid_dim | 64 |
| encoder_hid_dim_type | concat_char_embeds |
| in_channels | 1 |
| mid_block_type | UNetMidBlock2DCrossAttn |
| num_class_embeds | 2 |
| num_unique_chars | 11 |
| out_channels | 1 |
| up_block_types | [ CrossAttnUpBlock2D × 3, UpBlock2D ] |

## C  OPTIMAL SEQUENCE LENGTH AND IMAGE PARTS

The sequences in the vehicle type data vary in length, and we fix the sequence length by appending space characters. The fixed length and subsequent number of image parts to generate are important hyperparameters. Table 3 lists OCR accuracies resulting from different configurations of these two parameters. For each configuration we trained a diffusion model on the training set of the first split in Table 1 (set 1), then trained an OCR model on 10,000 images generated by the diffusion model,

Table 3: Performance of OCR models trained on synthetic data from different diffusion model configurations of fixed sequence length and number of image parts.

| SeqLen | Parts | Val. accuracy | Test accuracy |
|---|---|---|---|
| 5 | 2 | 84.75 | 89.27 |
| 5 | 3 | 76.95 | 96.19 |
| 6 | 2 | 81.98 | **99.05** |
| 6 | 3 | 66.55 | 95.35 |
| 7 | 2 | 95.49 | 98.69 |
| 7 | 3 | 88.56 | 98.57 |

and evaluated that OCR model on the validation and test sets of set 1. Based on these results, we pick the sequence length as 6 and the number of image parts as 2. These values are used in all the experiments and other dataset splits of Section 5.2.

# D    RESULTS IN TABLE FORM

## D.1    MOCK NUMBER PLATES

In Figure 5 we showed test accuracies from three OCR models trained on original, synthetic, and original plus synthetic data, for each of six different SADE training sets of mock number plate images. Table 4 lists these accuracies in table form.

Table 4: Test accuracies from models trained on original mock number plate data (OCR 1), synthetic data generated by SADE (OCR 2), and a combination of the original and synthetic data (OCR 3). The first column refers to the sequences used to train SADE.

| Training seq. | OCR 1 | OCR 2 | OCR 3 |
|---|---|---|---|
| 40 | 2.25 | 98.52 | 99.98 |
| 50 | 15.32 | 98.52 | 100.0 |
| 60 | 92.92 | 99.86 | 100.0 |
| 80 | 99.98 | 99.98 | 100.0 |
| 100 | 99.88 | 99.94 | 100.0 |
| No 5 in [6] | 99.96 | 99.98 | 99.98 |

We experimented further on the effects of the cumulative error that can occur from splitting an image into too many image parts, demonstrated in Figure 9. Using the set of 100 training sequences and splitting an image into seven parts with no overlap, an OCR model trained on synthetic data achieved an accuracy of 80.44%, while an OCR model trained on original plus synthetic data achieved an accuracy of 98.84%. Again using the 100 training sequences, but splitting an image into six parts with one character overlap, the accuracies for OCR models trained on synthetic data and on original plus synthetic were 92.54% and 99.80%, respectively.

## D.2    REAL-WORLD VEHICLE TYPE LABELS

In Figure 6 we showed test accuracies from three OCR models trained on original, synthetic, and original plus synthetic data, for each of five different SADE training sets of vehicle type label images. Table 5 lists these accuracies in table form, for each respective set's test images on the left and for the hold-out set (set 6 in Table 1) on the right. The pretrained model of Baek et al. (2019) achieved an accuracy of 43.40% on the hold-out set.

Table 5: Test accuracies from models trained on original real-world vehicle type data (OCR 1), synthetic data generated by SADE (OCR 2), and a combination of the original and synthetic data (OCR 3). The set numbers are those from Table 1, and the pretrained OCR model ("Pretr.") is from Baek et al. (2019).

| OCR accuracies on each set's test images | | | | | OCR accuracies on the hold-out set | | | |
|---|---|---|---|---|---|---|---|---|
| Set | OCR 1 | OCR 2 | OCR 3 | Pretr. | Set | OCR 1 | OCR 2 | OCR 3 |
| 1 | 7.73 | 99.05 | 99.55 | 0.00 | 1 | 84.91 | 37.74 | 88.68 |
| 2 | 7.41 | 87.65 | 100.0 | 50.00 | 2 | 94.34 | 83.02 | 96.23 |
| 3 | 72.22 | 91.67 | 100.0 | 19.44 | 3 | 90.57 | 96.23 | 94.34 |
| 4 | 50.00 | 43.64 | 95.45 | 9.09 | 4 | 73.58 | 77.36 | 100.0 |
| 5 | 52.30 | 22.37 | 93.75 | 33.55 | 5 | 50.94 | 56.60 | 94.34 |

