# OpenReview forum: "SADE: a Scene-text Autoregressive Diffusion Engine for Character Sequence Recognition"
_ICLR.cc/2025/Conference — ICLR 2025 Conference Withdrawn Submission_

### Official Review · Reviewer_MduD · 2024-10-29

**Soundness:** 2
**Presentation:** 2
**Contribution:** 2
**Rating:** 5
**Confidence:** 3

**Summary:**

This paper proposes SADE (Scene-text Autoregressive Diffusion Engine), which employs an autoregressive diffusion model OCR task to generate synthetic images, aiming to solve the problem of the generalization ability of OCR models in the presence of character location diversity and insufficient training data. This approach reduces the effective number of possible positions for a character and increases the diffusion model’s ability to generate characters in novel positions of the full sequence. The results indicate that SADE can indeed lead to substantial improvements in OCR accuracy in data-scare scenarios, particularly on sequences with characters at positions not encountered in the original training data.

**Strengths:**

1) Diffusion modeling to generate real images for controlled generation.
2) Segmentation of the image at different positions is used to realize the authenticity of the generated character images.
3) Designed for specific scenarios to improve OCR performance by generating synthetic data in the absence of data.

**Weaknesses:**

1) How to control the model to improve the quality of the generated data is not reflected.
2) Lack of comparison with traditional synthetic data methods and GAN-based methods.
3) There is no mention of the circumstances under which the quality of the generated images is poor and how to avoid it.

**Questions:**

1) Is it possible to provide an overall design flowchart for generating the model?
2) Can the methods used to generate the data be applied to other scenarios?
3) How do we evaluate the match between the generated characters and the background and the authenticity of the overall image?

---

### Official Review · Reviewer_zRfW · 2024-10-29

**Soundness:** 2
**Presentation:** 2
**Contribution:** 2
**Rating:** 5
**Confidence:** 4

**Summary:**

The authors claim that training an OCR model to recognize short character sequences in scenarios where labeled training image data and sequence diversity are limited will not perform well. This is because the characters to be recognized do not appear in all scene positions, and OCR models cannot recognize characters that do not appear in certain scene positions. Therefore, the authors explore training a diffusion model on a small number of labeled images to generate synthetic images with similar background statistics but new character sequences. To address this issue, the authors introduce SADE: a scene-text autoregressive diffusion engine that generates multiple image parts one by one, conditioned on previously generated image parts for visual coherency. This approach reduces the effective number of possible positions for a character and enhances the diffusion model's ability to generate characters in new positions within the full sequence.

**Strengths:**

The authors propose SADE: a scene-text autoregressive diffusion engine that cleverly segments existing limited labeled images and overlaps the segments appropriately. This allows the diffusion model to construct a dataset with more labeled training images and increased sequence diversity based on the limited labeled images available. Furthermore, it enables sequences to appear in positions that have never been seen before, thereby enriching the training data.

**Weaknesses:**

The paper is challenging to read due to the absence of illustrative images that could complement the text. Additionally, it fails to present a detailed process diagram explaining how SADE generates new images, and the experimental comparisons are not convincing. Furthermore, the related works are not well reviewed. There are many important recent text recognition methods are ignored by the authors, such as Textdiffuser, AnyText.

[1]Chen J, Huang Y, Lv T, et al. Textdiffuser: Diffusion models as text painters[J]. Advances in Neural Information Processing Systems, 2024, 36.

[2] Tuo Y, Xiang W, He J Y, et al. AnyText: Multilingual Visual Text Generation and Editing[C]//The Twelfth International Conference on Learning Representations.

**Questions:**

For a more intuitive understanding of the method, the paper should include a comprehensive pipeline for SADE. Additionally, the experimental results would be clearer if presented in tabular form, as the current use of bar charts without specific accuracy figures makes it difficult to compare the outcomes. To further illustrate the effectiveness of the SADE method, it would be advantageous to conduct comparative experiments that include a diffusion model trained without SADE for data generation. The OCR model could then be trained using this dataset, and its accuracy compared against the existing results, thereby providing a direct demonstration of SADE's effectiveness.

---

### Official Review · Reviewer_PcxF · 2024-10-29

**Soundness:** 2
**Presentation:** 2
**Contribution:** 1
**Rating:** 1
**Confidence:** 4

**Summary:**

The paper proposes a method for synthesizing scene-text training data using an autoregressive diffusion model. In contrast, a non-autoregressive model struggled to accurately place characters in novel positions. The proposed method is tested on two tasks: license plate recognition and vehicle trim label recognition.

**Strengths:**

The combination of autoregression with the diffusion generation process is innovative and effectively addresses the challenge. Traditional diffusion models trained on whole images tend to memorize character positions, making it difficult to generate characters in unseen positions. The proposed method, by generating image parts autoregressively, enhances diversity in output while maintaining visual coherence.

**Weaknesses:**

1. The proposed method is primarily evaluated on two highly specific tasks: license plate and vehicle trim label recognition. While these are relatively straightforward OCR problems, it remains uncertain how well the method will generalize to broader, more complex OCR scenarios.

2. The method currently generates images of fixed size, limiting its applicability to OCR tasks that require processing text of varying lengths.

3. The paper's experimental evaluation and ablation study are somewhat limited. For instance, a more comprehensive analysis of the generated images' adherence to the conditioned characters would be beneficial. Additionally, a comparison with non-autoregressive methods would provide insights into the method's relative performance. Figure 1 offers a preliminary glimpse, but further quantitative evaluation is necessary.

4. The paper would benefit from careful proofreading and editing. For example, the abstract should correct "data-scare scenarios" to "data-scarce scenarios."

**Questions:**

Would a traditional render-based synthesis engine suffice for the scenarios explored in the paper?

---

### Official Review · Reviewer_dBbg · 2024-11-04

**Soundness:** 2
**Presentation:** 2
**Contribution:** 2
**Rating:** 3
**Confidence:** 2

**Summary:**

The paper introduces a Scene-Text Autoregressive Diffusion Engine (SADE) designed to improve OCR for short alphanumeric sequences in scenarios with limited labelled training data. SADE generates synthetic images by progressively creating image parts conditioned on previously generated parts, enhancing the model's ability to recognize characters in new positions. Experiments with mock number plates and real-world vehicle type labels demonstrate that SADE improves OCR accuracy, particularly for sequences with characters in positions not seen during training.

**Strengths:**

1. The paper presents an innovative approach of autoregressive diffusion model for generating synthetic scene-text images, which is a creative solution to improve OCR performance in data-scarce scenarios.
2. The paper provides a comprehensive explanation of the SADE model, including the process of conditioning on character sequences, generating overlapping image parts, and the training and sampling procedures.
3. The authors conduct good experiments with both synthetic and real-world datasets, showcasing the practical benefits and robustness of their approach in enhancing OCR performance.

**Weaknesses:**

1. Paper lacks ablation studies of each component of the SADE model, such as the autoregressive approach, overlapping image parts, and character sequence conditioning which may help in understanding the necessity and impact of different parts of the model.
2. The paper might lack sufficient qualitative comparisons with other state-of-the-art OCR methods. Including visual examples and case studies would provide a clearer understanding of how SADE performs relative to existing technologies.
3. The focus on specific types of text sequences, such as vehicle type labels, might limit the generalizability of the findings. Testing SADE on a broader variety of text types and languages would strengthen the claims of its versatility.
4. The autoregressive nature of SADE, which involves generating image parts sequentially, can be computationally intensive and time-consuming compared to more straightforward deep learning models.

**Questions:**

1. Could you include ablation studies to demonstrate the impact of each component of the SADE model, such as the autoregressive approach, overlapping image parts, and character sequence conditioning?
2. Can you provide more qualitative comparisons, including visual examples, to illustrate how SADE performs relative to other state-of-the-art OCR methods in various scenarios?
3. Could you include a detailed error analysis to identify common failure cases of SADE and suggest potential improvements or future work to address these issues?
4. How well does SADE generalize to entirely new types of text sequences or different domains? Could you include experiments or discussions on the model's adaptability to diverse datasets?

---

### Author Response · Authors · 2024-11-25
**Submission withdraw**

We would like to express our gratitude to the reviewers for their time and effort in providing constructive feedback on our paper. These insights have been valuable for identifying areas for improvement. After careful consideration, we have decided to withdraw this submission to allow us to address these points and strengthen the quality of the work.

---

### Note · Authors · 2024-11-25

I have read and agree with the venue's withdrawal policy on behalf of myself and my co-authors.